# Gastrointestinal Tract Microbiome Effect and Role in Disease Development

**DOI:** 10.3390/diseases10030045

**Published:** 2022-07-08

**Authors:** Neira Crnčević, Mirsada Hukić, Sara Deumić, Amir Selimagić, Ada Dozić, Ismet Gavrankapetanović, Dženana Klepo, Monia Avdić

**Affiliations:** 1Department of Genetics and Bioengineering, International Burch University, Francuske revolucije bb, Ilidža, 71210 Sarajevo, Bosnia and Herzegovina; sara.deumic@ibu.edu.ba (S.D.); dz.h.klepo@gmail.com (D.K.); monia.avdic@ibu.edu.ba (M.A.); 2Academy of Sciences and Arts of Bosnia and Herzegovina, Center for Disease Control and Geohealth Studies, Bistrik 7, 71000 Sarajevo, Bosnia and Herzegovina; mirsadahukic@yahoo.com; 3Institute for Biomedical Diagnostics and Research Nalaz, Čekaluša 69, 71000 Sarajevo, Bosnia and Herzegovina; 4Department of Gastroenterohepatology, General Hospital “Prim. dr. Abdulah Nakas”, 71000 Sarajevo, Bosnia and Herzegovina; selimagic.amir@gmail.com; 5Department of Internal Medicine, General Hospital “Prim. dr. Abdulah Nakas”, 71000 Sarajevo, Bosnia and Herzegovina; ada.dozic@gmail.com; 6University Clinical Center Sarajevo, Clinic of Orthopedics and Traumatology, Bolnička 25, 71000 Sarajevo, Bosnia and Herzegovina; ismetgav@icloud.com

**Keywords:** gastrointestinal tract microbiota, small intestine cancer, celiac disease, Crohn’s disease, autism spectrum disorder, quorum sensing

## Abstract

In recent years, it has been shown that gastrointestinal microflora has a substantial impact on the development of a large number of chronic diseases. The imbalance in the number or type of microbes in the gastrointestinal tract can lead to diseases and conditions, including autism spectrum disorder, celiac disease, Crohn’s disease, diabetes, and small bowel cancers. This can occur as a result of genetics, alcohol, tobacco, chemotherapeutics, cytostatics, as well as antibiotic overuse. Due to this, essential taxa can be lost, and the host’s metabolism can be severely affected. A less known condition called small intestine bacterial overgrowth (SIBO) can be seen in patients who suffer from hypochlorhydria and small intestine cancers. It is characterized as a state in which the bacterial population in the small intestine exceeds 10^5^–10^6^ organisms/mL. The latest examination methods such as double-balloon enteroscopy and wireless capsule endoscopy have the potential to increase the accuracy and precision of diagnosis and provide better patient care. This review paper aims to summarize the effect of the gastrointestinal environment on chronic disease severity and the development of cancers.

## 1. Introduction

The dynamic small bowel environment is not very suitable for microbial colonies due to digestive enzymes and bile, short transit time, and food ingestion. Because of this, its microbial life is less diverse and needs to adjust and respond quickly to different luminal conditions. The bacteria present in this environment have lower biomass, and the number of bacterial colonies changes in various locations in the small bowel. The bacterial population in the duodenum is approximately 10^4^–10^5^ CFU/mL. From the duodenum, this number increases gradually to 10^7^–10^8^ CFU/mL in the ileum. Moreover, the bacteria present also changes from the proximal-distal segments of the small bowel and the colon. There is an increase in the proportion of Gram-positive to Gram-negative bacteria. Moreover, the balance of strict anaerobic and facultative anaerobic bacteria increases in this direction. The genera usually found in the small bowel include *Staphylococcus*, *Lactobacillus*, *Bacteroides*, *Clostridium*, and *Streptococcus*, and their presence is affected by aging and diet [1]. Due to aging, the conditions such as diabetes, pancreatic disease, and cancer have potential adverse effects on the formation and function of the small bowel. The aging enteric nervous system may cause selective neurodegeneration contributing to the GI symptoms such as dysphagia, GI reflux, and constipation [2]. The bacterial composition and content of the gastrointestinal (GI) tract is seen in Table 1.

The importance of microbial life in the small bowel can be seen in its involvement in folate, vitamin K, and amino acid metabolism. This phenomenon was examined in an experiment in which the metabolism of germ-free and healthy mice was compared [3,4]. It was found that germ-free mice need supplementation with vitamin B, more specifically B12, folate, biotin, and vitamin K. These two vitamins are produced by bacterial genera including *Eubacterium*, and *Fusobacterium*, *Bacteroides*, and *Propionibacterium*. Moreover, another function of the resident small bowel bacteria represents the protection of the intestinal mucosa from invading pathogens, production of bacteriocins, and biofilm formation. This is achieved by preventing the pathogens from entering epithelial cells and is also known as the barrier effect [5].

The host diet plays a vital role in the small intestine environment. The primary biochemical process that occurs is the fermentation of carbohydrates. Compared to the fecal metagenome, the small intestinal metagenome has significantly more genes involved in the metabolism of carbohydrates. The main processes include lactate fermentation, the pentose phosphate pathway, and the sugar phosphotransferase systems. Studying the small intestine can be very challenging due to its inaccessibility. In experiments involving the human small intestine microbiota, specific invasive methods have to be used. The most common methods for studying the small intestine include nasoduodenal catheters and esophagoduodenogastroscopy. Because of this, it is difficult to find individuals who would participate in this kind of study [1].

Recently, molecular diagnostic techniques have started to become more popular. The microbial composition in the GI tract can also be determined by analyzing the 16s ribosomal RNA gene in bacteria. The findings from molecular diagnostic techniques suggest that there are approximately 200 bacterial strains. *Bacteroidetes* and *Firmicutes* are predominantly found in vertebrates. They represent more than 90% of all bacteria in the GI tract. Factors that affect gut microbiota diversity include genetics, diet, gender, and environment [7].

Many external factors affect the small intestine environment, including alcohol, tobacco, antibiotics, chemotherapeutics, cytostatics, and other ingested materials. This can result in various diseases and conditions such as gastric cancer, idiopathic inflammatory disorders, post-infectious syndromes, and squamous cell esophageal cancer. On the other hand, external factors such as food antigens can cause different diseases to develop, such as food allergies, eosinophilic esophagitis, and celiac disease [8].

Studying the small intestine environment possesses many difficulties because of its inaccessibility. Hence, most studies use samples obtained from sudden death victims, colonoscopies, and small intestine transplantations. The accuracy of this method is compromised since there is a risk of contamination from the colon. There is not enough clinical data about this topic that could help doctors find the optimal treatment for each patient. Recent diagnostic methods such as double-balloon enteroscopy and wireless capsule endoscopy represent a beginning of a new diagnostic era for minor intestine abnormalities with good precision and accuracy. Since the dynamic environment of the small intestine is involved in the functioning of the entire body, more research in this field will provide us with answers about the overall homeostasis of the host and the development of associated diseases [9].

The aim of this study was to analyze the gastrointestinal tract environment, its effect, and its role in disease development.

## 2. Impact of Antibiotic Resistance on Gastrointestinal Tract Environment

The spread of antibiotic resistance represents one of the major global health care concerns. The post-antibiotic era crisis is becoming more severe every day due to the overuse of antibiotics. The severity of this problem is seen as an increased rate of mortality among patients with bacterial infections [10]. Antibiotics severely affect the diversity of the gastrointestinal microbiome. This can lead to the loss of essential taxa and changes in the host’s metabolism. If this occurs, the antibiotic resistance in the remaining taxa is further stimulated [11].

Since antibiotic-susceptible bacteria are eradicated, the remaining antibiotic-resistant bacteria grow and multiply to take their place. Although the microbiome’s diversity is reduced, the overall amount of bacteria in the gastrointestinal tract can increase after the use of antibiotics. A study was conducted on patients to test the effect of broad-spectrum antibiotics on the overall microbial load. The results revealed that treatments with β-lactams for seven days resulted in an increased microbial load. This was concluded after the analysis of the patient’s fecal samples. Their microbial load was two-fold higher when compared to their negative controls. Furthermore, the ratio of *Bacteroidetes* to *Firmicutes* was also increased [12].

In addition, the overuse of antibiotics also affects the host’s immune system. In a study conducted to examine the effect of antibiotics on mice, it was shown that the overuse of antibiotics results in changes in gene expression and immune system regulation. The mice were given antibiotic treatments from birth. The incidence of type 1 diabetes was increased after using pulsed dosing in susceptible mice. Moreover, these mice were shown to have a lower relative level of anti-inflammatory T cells [13].

## 3. Quorum Sensing and Biofilms in the Gastrointestinal Tract

As cell-population density changes, specific bacterial populations can change their gene regulation. This process is known as quorum sensing. Autoinducers represent chemical signaling molecules that are produced in this process. Their concentration is proportionally increased as a function of cell density. A threshold stimulatory concentration of these molecules leads to changes in gene expression. Both Gram-positive and Gram-negative bacteria can communicate and regulate many physiological processes using quorum sensing. This includes conjugation, virulence, biofilm formation, motility, and symbiosis. Biofilms represent assemblages of bacterial colonies commonly present within aquatic environments. They form protective matrixes that adhere to many surfaces. In humans, they are usually associated with oral infections and with the presence of specific implantable medical devices such as joint prostheses and catheters. In Gram-positive bacteria, oligopeptides are used for communication as autoinducers. On the other hand, autoinducers for Gram-negative bacteria are acylated homoserine lactones [14].

Autoinducers and quorum sensing receptors found on bacterial surfaces bind together after reaching a certain threshold. Then, the receptors can obtain the gene binding domains, which leads to the regulation of many physiological processes. An autoinduction feedback loop is formed and causes the development of bacterial populations. The GI surfaces can contain biofilms made up of fewer cells arranged in smaller aggregates and clusters around mucin aggregates. When it comes to GI parts in which the overall diversity of the enteric microbiota is poor, biofilms can still be heterogeneous. They can have cells that exhibit different genotypes [15].

If mature biofilms are detected on previously healthy tissue in the GI tract, this could indicate the formation of a damaged gut. An increased amount of *E. coli* forming biofilms was an early sign of ulcerative colitis [15]. The harsh GI environment can be very challenging for bacterial growth due to intestinal bile and stomach acid. The GI microbiota is surprisingly very stable because of quorum sensing [14].

## 4. Production of Neurotransmitters in the Gastrointestinal Tract

Recent studies suggest a clear link between gut microbiota and brain activity. The gut microbiota regulates the cognitive functions in the brain. The microbes in the GI tract serve as mediators in the communication among the peripheral immune, metabolic, and central nervous systems (CNS). This is achieved due to the presence of the microbiota-gut-brain axis. The gut microbiota balance is very sensitive. Specific changes caused by external or internal factors can lead to the disruption of this balance. As a result of this, many disorders and diseases can occur. These imbalances are seen in many neurological diseases such as epilepsy, Alzheimer’s disease, autism, Parkinson’s disease, and depression. In Alzheimer’s disease, patients possess an imbalanced gut microbiota. They often suffer from a lower abundance of *Bifidobacterium*, *Eubacterium rectale*, and *Dialister*. Their fecal microbial diversity is also lower when compared to healthy individuals. Furthermore, these patients contain a higher abundance of *Ruminococcus*, *Escherichia*, and *Bacteroides*. The gut-brain axis represents a network composed of the endocrine, central nervous, and immune systems. The information between the gut and the brain can be transmitted in both directions. Recent evidence shows that the gut microbiota can produce metabolites such as neurotransmitters and short-chain fatty acids. The levels of related metabolites present in the brain are affected by the functioning of the gut microbiota. This is possible because of the blood circulation and the blood-brain barrier. This way, the gut microbiota can regulate cognitive processes in the brain. Some studies suggest that the microbiota can produce lipopolysaccharides, which can trigger the peripheral immune system and cause the release of cytokines. This can cause the peripheral immune cells to pass into the brain and cause inflammation of the CNS [16]. However, this still remains a controversial topic, and more research needs to be conducted in order to confirm this.

A seminal study in 2004 used germ-free mice to examine if there is a connection between gut microbiota and the brain’s cognitive functions [17]. It was proven that germ-free mice had an increased response to stress. This was performed by using the restraint model. It was found that the behavioral changes caused by the lack of bacteria can be reversed by recolonizing the mice with a complete microbiota. This could be achieved by recolonization with *Bifidobacterium infantis* or a stool transplant [18].

Dopamine represents a catecholamine neurotransmitter that controls emotions, locomotor activity, food intake, endocrine regulation, and positive reinforcement. Additionally, it also functions as a modulator in the periphery for catecholamine release, hormone secretion, renal function, vascular function, and gastrointestinal motility [19].

Moreover, it is also a precursor for epinephrine and norepinephrine. For many years, it was thought that norepinephrine only plays a role in detecting sensory signals, alertness, and arousal in the waking state. However, recent studies show that it is also a part of the processes such as attention, learning, and memory. Some bacteria are also able to produce as well as respond to catecholamines. More specifically, the growth rate of pathogenic *Escherichia coli* O157:H7 (EHEC) increases in the presence of norepinephrine and dopamine. In the presence of norepinephrine, the rate of its biofilm formation, virulence, and motility is also shown to increase. Some other bacteria were also shown to change their behavior in the presence of catecholamines. Pathogenic *Shigella sonnei*, *Pseudomonas aeruginosa*, *Staphylococcus aureus*, *Klebsiella pneumoniae*, and *Enterobacter cloacae* increase their growth rate in the presence of norepinephrine. The change acquisition of iron could explain this. Norepinephrine is also present in relatively high amounts in the biomass of certain bacterial species. For instance, species such as *Bacillus mycoides*, *Proteus vulgaris*, *Bacillus subtilis*, *Escherichia coli*, and *Serratia marcescens* contain relatively high levels of norepinephrine. It is believed this molecule is produced in the quorum sensing processes [18].

## 5. Small Intestine Bacterial Overgrowth (SIBO) in Disease Development

Small intestine bacterial overgrowth, also known as SIBO, was once known as a disease present only in a small number of patients. However, new diagnostic methods for examining the small bowel bacterial population have enabled patients to receive a correct diagnosis and seek treatment immediately. These patients were usually misdiagnosed due to the wide range of nonspecific symptoms of SIBO, including abdominal pain, bloating, discomfort, fatigue, and diarrhea, which resembled irritable bowel syndrome, a far more common disease [20].

SIBO represents a state in which the small bowel contains a bacterial population that exceeds 10^5^–10^6^ organisms/mL. Two main factors can lead to SIBO. The first one is hypochlorhydria, abnormally low gastric acid secretion. Gastric acid is unable to limit the growth of the ingested bacteria. This is seen in patients diagnosed with *Helicobacter pylori* and older patients, those with high-stress levels, smokers, and excessive alcohol consumers. It is also seen in cases with a lack of nutrients such as iron, zinc, and vitamin B and long-term use of antacids and medicines for treating stomach ulcers or heartburn. The second factor represents small bowel dysmotility, which is seen in patients with abnormalities in the migrating motor complex. The primary function of the migrating motor complex is to clear the residual debris present in the GI tract while the person is fasting. In addition, the chance of developing SIBO is also increased if the GI tract has anatomical abnormalities. Certain GI tract surgeries, such as the Billroth II procedure, which results in a blind loop, can also lead to SIBO. This is because the residual secretions and food cannot be swept away effectively. Moreover, immunodeficient patients have a higher chance of developing SIBO [20].

Recently, an increased prevalence of SIBO has been noticed in patients who suffer from irritable bowel syndrome. Despite that, these studies included the use of positive early glucose breath tests, suggesting that it is an indirect way of diagnosing SIBO. Many disagree with this statement since there is a possibility that the findings from the glucose breath test include only the transit bacteria in the GI tract and not SIBO. In order to validate the results, quantitative cultures of the proximal small intestine have to be performed. Moreover, it is not possible to determine the type of bacteria of the SIBO with the use of glucose breath tests [21].

Another correlation between gut abnormalities and SIBO can be seen with cystic fibrosis. In a study by Lisowska et al., it was noticed that there is a significantly higher prevalence of SIBO in patients who have cystic fibrosis (CF) when compared to their healthy negative controls. The majority for SIBO was 37%, and the prevalence for the non-CF subjects was 13%. In order to diagnose SIBO in these patients, the hydrogen breath test (HBT) was used [22].

In experiments with mice, a higher prevalence of SIBO was found in mice with cystic fibrosis when compared to wild-type mice. There are two possible explanations for this outcome. Firstly, the accumulation of mucus due to cystic fibrosis could have acted as a bacterial anchor, which would have triggered their overgrowth. Secondly, the presence of unabsorbed lipids resulted in a slower intestinal transit. As a result, smooth muscle dysfunction and the ileal brake would have occurred. Because of this, it was concluded that the main symptoms of cystic fibrosis, including abnormal mucus clearance and gut dysmotility in patients, increase their chance of developing SIBO [6].

Recent evidence suggests that there is a correlation between SIBO and stunted children. In a study conducted by Vonaesch et al. [23], children from Bangui, Central African Republic, between the ages of 2 and 5 were examined for small intestine bacterial overgrowth. Gastric, duodenal, and fecal samples were collected from stunted and non-stunted children. This study revealed that small intestine bacterial overgrowth was seen in stunted children. The most common bacteria in these children were the ones that usually reside in the oral cavity. The abundance of *Campylobacter species*, *Escherichia coli*, and *Shigella* in the enteric microbiome was also noticed in stunted children. This confirmed that small intestinal bacterial overgrowth is often present in chronically malnourished children. Another study conducted by Donowitz et al. [24] also analyzed whether small intestine bacterial overgrowth contributed to stunted growth in children. The study included 90 children from Bangladesh who were two years old. It was shown that approximately 16.7% of them suffered from SIBO. The two main predictors for having SIBO were the proximity of an open sewer to children’s homes and stunting. When compared to their negative controls, children who suffered from SIBO possessed elevated fecal Reg 1β, fecal calprotectin, and intestinal inflammation markers. Because of this, it was concluded that small intestine bacterial overgrowth was linked to elevated signs of inflammation of the small bowel and stunting [25].

When it comes to therapy, two approaches can be used to treat SIBO. The first one is the treatment of each predisposing condition separately. The second one is the use of antibiotics. There are many difficulties surrounding the use of antibiotics for SIBO due to the fact that a large number of bacterial species are involved. Since they all have different antibiotic sensitivity, choosing antibiotic therapy can be a difficult task. In the past, tetracyclines were widely used. However, because of their low eradication rate of approximately 30% and many side effects, they have become a less popular choice in the treatment of SIBO. The low eradication rate is explained by their inability to target anaerobes directly. Instead of tetracyclines, patients have started using metronidazole [26].

Recently, studies that examine the effect of rifaximin on SIBO were conducted. Patients were given 1200 mg of rifaximin per day for seven days. The results suggested that this antibiotic was effective in fighting against SIBO. It was also found more effective than metronidazole [27]. In a study conducted by Attar et al., the efficacy of treating SIBO- diarrhea by norfloxacin, amoxicillin-clavulanic acid, and *Saccharomyces boulardii* was compared. This study included ten patients with SIBO diarrhea. It was concluded that using amoxicillin-clavulanic acid and norfloxacin is an effective way of treating SIBO diarrhea [28].

SIBO is characterized as relapsing because after receiving the first successful dose of antibiotics, approximately 44% of patients suffer from the same symptoms again after nine months. Based on the severity of the symptoms and their rapidity of return, patients usually need to follow a regimen of rotating different antibiotics. This rotation is followed for 1 to 2 weeks each month. Once in a while, some patients need to follow a continuous regimen of antibiotics. However, this kind of treatment has not been studied enough. In most cases, the patients are tested for SIBO before the new antibiotic therapy [29].

Most studies involving the analysis of SIBO in children included subjects with irritable bowel syndrome, chronic abdominal pain, those who live in poor conditions, obese children, etc. These studies’ significant difficulties represent a lack of appropriate controls and difficulty reaching the small bowel microbiota without contaminating the samples [6]. The major findings of small intestine bacterial overgrowth are seen in Table 2.

### 5.1. Celiac Disease and the Gut Microbiota

It is estimated that the prevalence of celiac disease is between 1% and 2% in the general population. Many patients often do not receive an accurate diagnosis due to the subtle and nonspecific symptoms. The clinical manifestation of this disease can occur at any age [30].

Celiac disease is a digestive and immune disorder induced by gluten consumption. The gluten protein contains proline and glutamine. Humans are not able to digest gluten completely in their upper GI tract. A bulk of toxic components in gluten are stored in gliadin, an alcohol-soluble part of gluten. Since pancreatic, gastric and intestinal brush-border membrane proteases cannot degrade all of the gliadin molecules when a person ingests gluten, it is left in the intestinal lumen. A peptide from the α-gliadin fraction represents an undigested molecule of gliadin. It is composed of 33 amino acids. During digestion, the peptides go through the intestinal epithelial barrier. This usually occurs when the intestinal permeability increases or there is an intestinal infection. If this happens, the antigen-presenting cells and the peptides can interact in the lamina propria. When the immune response to the gliadin fraction occurs, patients who have celiac disease have an inflammatory reaction. In most cases, the inflammation occurs in the upper small bowel. The epithelium and the lamina propria become saturated with chronic inflammatory cells. Additionally, the intestinal villi disintegrate, leading to villous atrophy [31].

The most common symptoms of celiac disease include anemia, fatigue, diarrhea, and weight loss. In a study conducted by Sánchez et al. in 2013, 57 children with active celiac disease, nonactive celiac disease, and healthy ones were analyzed to assess whether their duodenal mucosal microbiota differed in biodiversity and composition from each other. The results indicated that the microbiota of patients with celiac disease has differences in the abundance and diversity of some cultivable bacterial taxa. In the active phase of celiac disease, patients possessed a higher quantity of members of *Enterobacteriaceae*, *Proteobacteria*, *Staphylococcaceae*, and *Proteobacteria*. Furthermore, a reduced abundance of *S. mutans* and *S. anginosus*, members of the *Streptococcus* family, was observed in patients with active and nonactive celiac disease. These differences were reduced after a long period of a gluten-free diet. However, the patient’s microbiota was not restored completely [32].

It is suggested that the genotype of infants at risk of having celiac disease influences the composition of the early gut microbiota. An increased amount of pathogenic bacteria in some of these infants was also found in their gut. However, the genetic aspect also influences the formation of the gut microbiota and has to be considered when diagnosing patients with celiac disease [33].

In addition to genetics, many environmental factors play a role in the gut microbiota composition in infants. For instance, during vaginal delivery, vertical transmission of the gut microbiome’s *Bifidobacteria* and *Bacteroides* from mother to child is established. On the other hand, infants born via a caesarean section exhibit fewer *Bacteroidetes*, so their microbiome is less diverse. Because of this, some believe that there is a correlation between infants born via a cesarean section and the development of celiac disease. However, this topic, for now, remains controversial [30].

### 5.2. Autism Spectrum Disorder and Gut Microbiota

It has become evident that there is a clear connection between the CNS and the host’s microbiome. Although the exact pathway is not known, it is now known that the enteric nervous system acts as a linkage between the CNS and GI microbiota. When the host is subjected to extreme conditions such as treatment with antibiotics or a germ-free environment, the levels of a large number of monoamine neurotransmitters, as well as neurotrophins, change. These molecules are needed for the proper development of the brain and its plasticity. Because of this, the connection between brain disorders and the GI gut microbiota is a popular topic of discussion in scientific circles [34].

Autism spectrum disorder represents a broad range of developmental brain disorders. A study conducted in 2012 showed that the approximate prevalence of this disorder was 14.6 per 1000 children. Furthermore, it was found that this prevalence was higher in males than in females. The estimated prevalence was 23.6 per 1000 in males and 5.3 per 1000 in females. Recent studies suggest that diarrhea, abdominal pain, flatulence, gaseousness, and constipation are frequently found in patients with an autism spectrum disorder. The prevalence of these kinds of symptoms is estimated to be between 23% and 70% in children who have autism spectrum disorder [35].

Recent evidence shows that the gut microbiota composition in children with autism spectrum disorders is significantly different compared to healthy children. Dysbiosis represents the microbial imbalance that occurs due to an increased occurrence of pathogenic bacteria. The mucosal barrier is disrupted. As a result, the intestinal permeability of neurotoxic peptides from bacteria and the exogenous peptides from food is increased. Inflammatory cytokines are produced. Because the gut-brain axis is disrupted, the host’s metabolic, endocrine, and neural mechanisms are disrupted. This is related to many neuropsychiatric disorders such as autism spectrum disorders. Because of this, specific changes in the gut microbiota are now known to be a risk factor in people who have a genetic predisposition to have autism spectrum disorders [36].

In dysbiosis, lipopolysaccharide (LPS), a pro-inflammatory endotoxin, is released into the bloodstream. It can then affect the processes in the CNS. In particular, the activity of parts of the brain used for emotional control is increased. In a study conducted by Emanuele et al. [37], LPS serum levels were measured in patients who have autism spectrum disorders. The results showed significantly higher LPS serum levels in patients with autism spectrum disorder [38]. Moreover, it was demonstrated that the enteric gut microflora of children with autism spectrum disorder is different from healthy controls and their neurotypical siblings. Their gut microbiome is characterized by a reduced microbial diversity and an increased microflora [36].

After analyzing the fecal samples of children with autism spectrum disorder, large amounts of *Clostridium* genus were found. The number of species under this genus was found to be ten times higher when compared to healthy controls. In addition, the analysis of the gut microbiota suggests an increased amount of Bacteroidetes, *Sutterella*, *Bifidobacterium*, *Prevotella*, *Alcaligenaceae* family, and *Ruminococcus genera* [38].

Fluorescent in situ hybridization-based techniques revealed that children affected by autism spectrum disorder possessed increased amounts of *Clostridia*. These children were also found to have an increased amount of *Bacteroides vulgatus* and *Desulfovibrio.* A correlation between the presence of these bacteria and the severity of the autism symptoms was also identified. In particular, a study conducted on Slovakian children who have autism spectrum disorder showed that they contained an increased amount of *Desulfovibrio* compared to their healthy negative controls. This Gram-negative bacterium could be one of the factors that cause inflammation due to the fact that it can reduce sulfates. This metabolic byproduct is very toxic to epithelium cells in the colon. Moreover, an experiment involving rodents showed that the presence of *Desulfovibrio* resulted in a decreased working memory. It was also found that an increased amount of *Sutterella* is present in a large number of biopsies from children with autism spectrum disorder who had specific GI abnormalities. The biopsies were taken from the intestine [39].

Another study analyzed 40 patients with autism spectrum disorder and their corresponding healthy negative controls. Thirty-six of these children had severe symptoms. The children with autism spectrum disorder contained a decreased amount of *Parabacteroides*, *Veillonella*, *Bilophila*, *Alistipes,* and *Dialister*. However, they also had increased *Lactobacillus*, *Dorea*, and *Collinsella*. This confirmed that their GI composition was significantly different. Moreover, symptoms such as constipation in children with autism spectrum disorder were linked to an increased amount of *Clostridium* and *Shigella*/*Escherichia* cluster XVIII. This cluster is responsible for producing a large number of exotoxins, which are known to be pro-inflammatory [39].

### 5.3. Crohn’s Disease and Gut Microbiota

Crohn’s disease is an inflammatory bowel disease characterized by abdominal pain, intestinal ulcers, diarrhea, and cramping symptoms. Some studies concluded that this disease is associated with microbial dysbiosis. This change in the microbial population of the GI tract can lead to inflammation. However, inflammation can also cause this change. Patients who suffer from inflammatory bowel syndrome usually receive large amounts of antibiotics before they are diagnosed to alleviate the symptoms. This can cause drastic changes in the GI microbiota. In a study conducted by Xavier et al., biopsies of the lower part of the rectum and small intestine and fecal samples were taken from 447 children. These children were diagnosed with Crohn’s disease. The control group contained 221 children who suffered from non-inflammatory abdominal symptoms. Most of these patients did not use anti-inflammatory drugs and antibiotics at the time. The results showed that patients with Crohn’s disease contained more *Enterobacteriaceae*, *Veillonellaceae*, *Pasteurellaceae*, and *Fusobacteriaceae*. On the other hand, these patients had lower amounts of *Clostridiales*, *Bacteroidales*, and *Erysipelotrichales* [40].

The impact of the gut microbiota on the pathogenesis of Crohn’s disease is seen in the fact that inflammation usually occurs in places with the most significant amount of bacteria in the GI tract. This kind of inflammation can be reduced with the help of antibiotics and an elemental diet. Moreover, it was found that the small intestine tissue of patients who suffer from Crohn’s disease is incapable of killing. Animal models of inflammatory bowel disease enable scientists to understand better the impact of antibiotics on the GI environment and the extent of inflammation. These studies showed that in models of Crohn’s disease, inflammation is present in a non-sterile environment [41].

The resident microflora in the GI tract is called commensal microflora in healthy individuals. This term represents the symbiotic relationship that is present between two microbes. In this relationship, one microbe has certain benefits. The other microbe is not harmed, but it also does not have any help. The symbiotic relationship does not affect every microbe in the GI tract [42].

Several theories explain that one or more pathogenic microorganisms control inflammation. The most common of these is *Mycobacterium avium* subspecies *paratuberculosis* (MAP). This bacterium is frequently found in dairy food. Patients with Crohn’s disease usually have a higher amount of this bacterium in their biopsy and fecal samples when compared to their healthy negative controls. Another bacterium found in greater amounts in Crohn’s disease patients is *E. coli* LF82. However, this topic remains controversial since the increased levels of these bacteria can also occur due to inflammation and not its actual cause [43].

In addition to bacteria, it is also believed that fungi play an essential role in developing Crohn’s disease. In a study conducted by Chehoud et al., it was found that patients with Crohn’s disease possessed a unique fungal environment and a less diverse bacterial community. It was shown that patients with Crohn’s disease contained a greater abundance of two lineages of *Candida* when compared to their healthy controls. The *p* values of the Crohn’s disease patients for these lineages were 0.0034 and 0.00038. On the other hand, the healthy negative controls contained a greater abundance of *Cladosporium*. This *p*-value was 0.0025. Furthermore, it was noticed that the abundance of archaea in both groups was not significantly different [44].

In order to achieve the highest possible GI balance, Crohn’s disease usually receives a combination of probiotics, prebiotics, and symbiotics. Furthermore, patients need to follow a strict diet that can help reduce this disease’s symptoms. Moreover, fecal microbiota transplantation (FMT) can help Crohn’s disease patients to relieve the symptoms. It is estimated that 50.5% of patients have fewer symptoms after transplantation. However, this is still a controversial idea, and more research needs to be conducted. The most common diet for Crohn’s disease patients is the low intake of fermentable oligosaccharides, disaccharides, monosaccharides, and polyols, known as the FODMAP diet. This diet is rigorous and requires the patient to exclude polyols, fermentable oligosaccharides, monosaccharides, and disaccharides. The intestinal polyols include xylitol, fructose, and lactose. This diet is usually suggested because of its ability to reduce symptoms such as bloating and pain. It drastically improves patients’ quality of life with severe symptoms [45].

### 5.4. Diabetes and Gut Microbiota

In recent years, the prevalence of diabetes has never been higher due to unhealthy lifestyles and obesity. In 2019, with an estimated 1.5 million deaths, diabetes was the ninth leading cause of death. It was also estimated that by 2035, the number of people with diabetes would reach 592 million. More than 85% of the total prevalence corresponds to type 2 diabetes. This disease is very dangerous because it can have various symptoms, including nephropathy, neuropathy, retinopathy, peripheral vascular disorders, strokes, and ischaemic heart disease [46].

There is increasing evidence that there is a link between SIBO and diabetes. The incidence of SIBO in patients with diabetes is higher when compared to healthy patients. This is most frequently seen in patients with type 2 diabetes who also suffer from peripheral neuropathy. Moreover, patients with SIBO and type 2 diabetes have a higher chance of developing severe diabetic complications. These patients often have many GI complications caused by an imbalance of the gut microbiome. In a study conducted by Guan et al., out of all patients with type 2 diabetes in the experiment, 53.85% were SIBO positive [47].

When it comes to type 1 diabetes, patients usually experience micro- and macro-vascular complications. The most common complications of type 1 include nephropathy, retinopathy, and neuropathy. More specifically, patients often have diabetic autonomic neuropathy (DAN). This symptom is seen in approximately 20% of patients. It can cause GI disturbances such as diarrhea, esophageal enteropathy, constipation, and gastroparesis. If the intestinal motility and gastric emptying are disturbed, this can trigger the SIBO to form in the GI tract [48].

Type 2 diabetes patients have a decreased abundance of *Firmicutes*, *Bifidobacteria*, and *Clostridium* in their GI tract. However, they also have an increased abundance of β-proteus as well as *Bacteroides*. A positive correlation between the ratios of *Firmicutes/Clostridium* and *Bacteroides/Firmicutes* with blood glucose levels was also noticed. These ratios were not dependent on the patient’s weight. The analysis of fecal samples of patients with diabetes showed that their *Bacillus* content was lower than usual. Furthermore, the levels of *Bifidobacteria* present in the small intestine were lower when compared to healthy individuals. Interestingly, the abundance of *Enterococcus* fecal was also higher [49].

## 6. Small Bowel Cancers

Small bowel cancers are very rare, and their global incidence is less than 1.0 per 100,000 population. They represent 0.42% of the total cancer cases worldwide and can be divided into four types: lymphomas, adenocarcinomas, sarcomas, and carcinoid tumors. Recently, these cancers have become more prevalent. More specifically, a four-fold increase was observed for carcinoid tumors. This was especially seen in individuals with celiac disease, Peutz-Jeghers syndrome, familial adenomatous polyposis, and Crohn’s disease. It is also believed that obesity, smoking, and a diet filled with smoked or red meat also increase the risk of small bowel cancers [50].

Many believe that such a low incidence of small bowel cancer can be explained by its increased rate of apoptosis, rapid transit time, the relative absence of bacteria in a healthy small bowel, and the IgA-mediated immune system. Firstly, it is thought that the partially transformed cells are removed from the GI tract before complete carcinogenesis can happen. Secondly, it was found that some carcinogens induce colon cancer in healthy animals. However, colon cancer is not caused in germ-free animals. Certain products of bacterial breakdown can be carcinogens. Since there is a relative absence of bacteria in the small bowel, the exposure to potential carcinogens is minimized. Thirdly, the small bowel possesses enzymes that can act as protection against carcinogenesis. Benzopyrene is a substance that can act as a potent carcinogen in the small bowel [51].

When it comes to small bowel adenocarcinomas, patients usually have no symptoms at first. Then, nonspecific symptoms start to appear. In many cases, patients are thought to have irritable bowel syndrome [51]. Symptoms include abdominal pain, obstruction, nausea, vomiting, and fecal occult blood. Obstruction can occur when the bowel lumen is narrowed down due to the presence of the tumor. An endoscopy can be performed to confirm the presence of lesions in the proximal jejunum and duodenum. Radiographic imaging can also be used in some cases. However, if the patient possesses lesions distal to the Treitz ligament, endoscopy is not the most suitable method. This is because of the length of the small bowel. In that case, doctors can use capsule endoscopy, which represents an effective and less painful way of diagnosing small bowel tumors [52]. Small bowel sarcomas occur due to the transformed cells, which have a mesenchymal origin. At first, the patients are asymptomatic or have abdominal pain. In 66% of cases with small bowel sarcomas, a palpable abdominal mass is formed. These sarcomas are usually aggressive, and patients have a poor prognosis [53].

In the GI tract, infections with Helicobacter pylori represent the most critical risk factors for gastric adenocarcinomas. This is seen in 1 to 2% of patients infected with this pathogen [54]. Adenocarcinomas appear as a proliferation of the mucosal epithelial cells. They are found on the small intestine linings and are present as benign polyps at first. However, they can become immortalized. If this occurs, they can develop into adenocarcinomas, which usually happen after a latent period between 10 and 20 years. Their precursors, adenomas, represent polyps of granular cells. It is known that 10% of all adenomas develop into adenocarcinomas in the large intestine. However, this proportion is not known for the small intestine adenocarcinomas because there are no regular screening procedures and its overall rarity. The density of the microbiota, which can produce anti-cancerogenic deoxycholic acid from bile salts, instantly reduces the rate of these cancers in the human GI tract. Moreover, the impact of dietary carcinogens is lowered because of the shorter transit time through this organ. When it comes to the small intestine adenocarcinomas, they are usually found in the duodenum. This is where pancreatic juice and bile are released in indigestion [55].

It is tough to guide patients with adenocarcinomas because of the limited data about this condition. Many patients receive a delayed diagnosis. In most cases, this usually results in the development of diseases such as distant metastatic disease. In the past, examining the whole small intestine possessed many changes. For many years, doctors have used barium small bowel intestine follow-through as their primary way of diagnosing this cancer. This method included a 60% sensitivity in patients with an advanced stage of small intestine tumors. Additionally, cross-sectional imaging with magnetic resonance imaging (MRI) or computed tomography (CT) can be used to examine distal metastatic diseases or local nodal involvements. However, this method cannot successfully pinpoint primary lesions. The diagnostic sensitivity can be improved by using enteroclysis. The small intestine is directly infused with a contrast material using a nasogastric tube. Instead of this, a neutral contrast agent can also be used to improve the diagnostics method’s sensitivity. These methods are not used very often in hospitals. It is impossible to thoroughly examine the small intestine by endoscopy due to its length of up to five meters. If push enteroscopy is performed long enteroscope is used, enabling the doctor to examine the proximal 150 to 200 cm of the small intestine. A double-balloon enteroscopy is a relatively new method of studying the small intestine. However, this method is very time-consuming and is only performed under some strict conditions at specialized centers. One of the most modern ways of diagnosing small intestine cancers represents using wireless capsule endoscopy. In 2001, it was first approved in the United States to analyze the small intestine’s lumen. It was mainly used for identifying GI bleedings [56].

## 7. Conclusions

The composition of the gastrointestinal microbiota has a strong impact on the homeostasis of the entire organism. Imbalances of this part of the body can lead to the occurrence of many conditions and diseases, including autism spectrum disorder, celiac disease, Crohn’s disease, diabetes, and small bowel cancers. Specifically, the imbalance that causes these conditions usually occurs in the small intestine, when a condition called small intestine bacterial overgrowth develops. The homeostatic imbalance is especially seen after the overuse of antibiotics when the microbiome’s diversity becomes reduced, and the host’s immune system becomes severely affected. Moreover, the importance of the gut microbiome is seen in its effect on cognitive functions because of its role in regulating reactions in the brain. Since the gut microbiome environment is involved in the functioning of the entire body, more research in this field will provide us with answers about the overall homeostasis of the host and the development of associated diseases.

## Figures and Tables

**Table 1 diseases-10-00045-t001:** GI tract bacterial count and composition in humans. Data from [6].

GI Tract	Bacterial Count	Bacterial Composition
Stomach	10^3^–10^4^/mL content	*Streptococcus* *Staphylococcus* *Enterococcus* *Lactobacillus*
Duodenum/jejunum	10^3^–10^4^/mL content	*Corynebacterium*
Ileum	10^8^/mL content	*Escherichia coli* *Klebsiella* *Enterobacter* *Bacteroides* *Eubacterium* *Clostridium* *Ruminococcus*
Colon	10^11^/mL content	*Bifidobacterium*

**Table 2 diseases-10-00045-t002:** Small intestine bacterial overgrowth major findings. Data from [20,22,23,24,26].

Small Intestine Bacterial Overgrowth	
Symptoms	abdominal pain, bloating, discomfort, fatigue, and diarrhea
Factors that lead to SIBO	hypochlorhydria, small bowel dysmotility, anatomical abnormalities of the GI tract, GI tract surgeries, immunodeficient patients, irritable bowel patients, cystic fibrosis patients, stunted growth
Treatment	treatment of each predisposing condition separately antibiotics

## Data Availability

Not applicable.

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
