# Peer review of "Gastrointestinal Tract Microbiome Effect and Role in Disease Development"

_diseases, 2022, doi:10.3390/diseases10030045_

Round 1

Reviewer 1 Report

Having been excited by the title of this paper I was disappointed by much of the content.  The title is 'small bowel bacterial overgrowth inflammation and cancer'.  However large sections of this report have no relevance to small intestinal bacterial overgrowth - including the sections on Helicobacter pylori (I was waiting for revelations on small intestinal disease due to this organism!), Crohn's disease and autism which appear to touch only on faecal/colonic flora and do not include any relevant information on the small intestine and whilst there is existing data on small intestinal microbiome in coeliac disease, the authors appear to have only referenced the colonic microbiome studies, but have not made this clear.

It is common when reviewing such reviews on the small intestinal microbiome for the authors to fail to distinguish between the colonic and the small intestinal microbiome, and this is entirely evident in this submission.  Given the title I would expect the authors to focus entirely on small intestinal microbiome, and also to clarify the methods that were used to study the microbiome as this can impact on the relevance of the findings (even in the colon - whether on biopsy or stool sampling for instance).

Similarly in articles of this nature, authors often fail to attempt to try to dissect cause and effect from association - in many instances the direction of cause and effect (such as with the association of bacterial overgrowth and childhood stunting) could be at the level of association with a different factor (childhood poverty/environment), or either direction (stunting causing overgrowth or overgrowth causing stunting).  This is just one example of lack of lack of detail or clarity in an article that superficially appears to cover pathogenesis, presentation and treatment of conditions as diverse as gastric and intestinal cancer, autism, coeliac disease, Crohn's disease etc - the article has gone way off message and by the end does not really appear to bear any relevance to the title.

There are some statements that I find concerning - such as peripheral immune cells infiltrating the central nervous system (this is unlikely and the authors appear to suggest that this is somehow driven by specific gut microflora), and the comment about the IGA immune system being a reason why small intestinal cancers are less common than colonic cancers - although the IgA immune system is present and extremely active in the colon....

The lack of detail, the degree of extrapolation of interesting research findings without clarification or support, the wandering off message, the lack of discrimination of small intestinal and colonic flora effects, amongst other reasons were disappointing after my initial excitement.  Some sections were genuinely interesting - such as the paragraphs on quorum sensing and biofilms - but were inadequately expanded into points of relevance and could not save the overall poor quality of this review.

Author Response

Point 1: Having been excited by the title of this paper I was disappointed by much of the content. The title
is 'small bowel bacterial overgrowth inflammation and cancer'. However large sections of this report have
no relevance to small intestinal bacterial overgrowth - including the sections on Helicobacter pylori (I was
waiting for revelations on small intestinal disease due to this organism!), Crohn's disease and autism which
appear to touch only on faecal/colonic flora and do not include any relevant information on the small
intestine and whilst there is existing data on small intestinal microbiome in coeliac disease, the authors
appear to have only referenced the colonic microbiome studies, but have not made this clear.
Response 1: We completely agree with this comment and believe that by changing the title, this issue is
solved.
Point 2: It is common when reviewing such reviews on the small intestinal microbiome for the authors to
fail to distinguish between the colonic and the small intestinal microbiome, and this is entirely evident in
this submission. Given the title I would expect the authors to focus entirely on small intestinal microbiome,
and also to clarify the methods that were used to study the microbiome as this can impact on the relevance
of the findings (even in the colon - whether on biopsy or stool sampling for instance).
Response 2: Just like in the previous response, the title change should solve most of these problems. We
tried to include as many disease associations as possible in order to present the impact of the
gastrointestinal environment on disease association and development. Because of this, the information
seems to lack detail.
Point 3: Similarly in articles of this nature, authors often fail to attempt to try to dissect cause and effect
from association - in many instances the direction of cause and effect (such as with the association of
bacterial overgrowth and childhood stunting) could be at the level of association with a different factor
(childhood poverty/environment), or either direction (stunting causing overgrowth or overgrowth causing
stunting). This is just one example of lack of lack of detail or clarity in an article that superficially appears
to cover pathogenesis, presentation and treatment of conditions as diverse as gastric and intestinal cancer,
autism, coeliac disease, Crohn's disease etc - the article has gone way off message and by the end does not
really appear to bear any relevance to the title.
Response 3: By changing the title, the information in the article should become more relevant.
Point 4: There are some statements that I find concerning - such as peripheral immune cells infiltrating the
central nervous system (this is unlikely and the authors appear to suggest that this is somehow driven by
specific gut microflora), and the comment about the IGA immune system being a reason why small
intestinal cancers are less common than colonic cancers - although the IgA immune system is present and
extremely active in the colon....
Response 4: There is a paper that talks about this and suggests this correlation. In the updated version of
our paper, it is stated that this information is still controversial and more research needs to be done in order
to confirm this.
Point 5: The lack of detail, the degree of extrapolation of interesting research findings without clarification
or support, the wandering off message, the lack of discrimination of small intestinal and colonic flora
effects, amongst other reasons were disappointing after my initial excitement. Some sections were
genuinely interesting - such as the paragraphs on quorum sensing and biofilms - but were inadequately
expanded into points of relevance and could not save the overall poor quality of this review.
Response 5: We hope by changing the title and acknowledging Your suggestions, the paper seems more
relevant and the overall quality is improved. The new aim of this paper is to analyze the gastrointestinal
tract environment, its effect, and role in disease development and association.

Reviewer 2 Report

Diseases-1721881

Title: Small Bowell Bacterial Overgrowth Inflammation and Cancer

Brief abstract

This review was focused to describe foundations about the role of microbiota dysbiosis caused by several factors (antibiotics, neurotransmitters, and so on) that favored the small intestine bacterial overgrowth (SIBO) and its impact in the development and severity of chronic illness including celiac disease, Crohn’s disease, autism spectrum disorder, diabetes, and intestine cancer.

Highlights

This review provides a very interesting topic of small intestine bacterial overgrowth given that underlying mechanisms that disturb small bowel homeostasis have been scarcely documented. The review provides foundations with potential impact in diagnosis and therapeutic interventions of SIBO and the outcome of several illnesses of clinical relevance including celiac disease, Crohn´s disease, small bowel cancer among others. Congratulations for all authors

Major concerns

1. The content of this manuscript should be re-organized

2. Readably of the whole text may be improved to avoid “gaps” in the transition of each statement and/or paragraph and to change enunciates and/or paragraph at the right place

3. Title should be changed to reflect the content of this review regarding the implications of SIBO in diseases of clinical relevance not just inflammation and cancer

4. Overall background stated in “Conclusion” should be placed as “Introduction”

5. Originality and the aim of this review should be explicitly stated at the end of “Introduction”

6. In the current version “Introduction” should be an additional a section entitled for example “Small bowel microbiota”

7. From my point of view, this review is not a “Metadata Analysis” conducted according the Prisma guidelines; the manuscript is a “Descriptive Review”; anyhow, authors should enquire the author´s guidelines of MDPI Journal “Diseases” to clarify whether the section of “Material & Methods” has to be included as seen in the current version.

8. Although in the title “3. Quorum sensing and biofilms…” indicates “in the small bowel” the content of this section refers only “Gastrointestinal tract” so this should be clarified please.

9. In the section “4. Production of neurotransmitters by gut microbiota” clarify please whether is in the “Gastrointestinal tract” or specifically in “Small bowel”

10. it would be very valuable for the readers to include a Table in the section “5. Small intestine Bacterial Overgrowth…” that recapitulates main findings of experimental assays and studies based on clinical trials and potential mechanisms underlie SIBO

11. it would be very valuable for the readers to include a Figure that depicts mechanisms underlie SIBO based on the aforementioned Table.

12. In the sections “5.1 Celiac Disease and the gut microbiota”, “5.2 Autism spectrum disorder and gut microbiota” and “5.3 Crohn´s disease and gut microbiota” clarify please explicitly whether these diseases affect the small bowel or the whole gastrointestinal tract

13. The conclusion should highlight the impact of SIBO in diagnosis and potential therapeutic interventions of pharmaceutical and/immunologic type that reduce and/or replace the use of antibiotics

14. check out and correct please “Chron´s disease” spelling in the body text and key words

See please in attached PDF additional corrections

Author Response

Point 1: The content of this manuscript should be re-organized
Response 1: We re-organized some parts of the paper by following Your suggestions. Certain changes were
made to the introduction and conclusion. A part of the introduction talking about antibiotic resistance is
now a separate subtitle.
Point 2: Readably of the whole text may be improved to avoid “gaps” in the transition of each statement
and/or paragraph and to change enunciates and/or paragraph at the right place
Response 2: We moved certain paragraphs to new places in the paper. This should improve the overall
flow and readability of the paper. Furthermore, the title change should solve this issue.
Point 3: Title should be changed to reflect the content of this review regarding the implications of SIBO in
diseases of clinical relevance not just inflammation and cancer
Response 3: We completely agree with this statement. The title is now changed and should cover all of the
subtitles in the paper.
Point 4: Overall background stated in “Conclusion” should be placed as “Introduction”
Response 4: This now changed.
Point 5. Originality and the aim of this review should be explicitly stated at the end of “Introduction”
Response 5: This is now changed.
Point 6. In the current version “Introduction” should be an additional a section entitled for example “Small
bowel microbiota”
Response 6: The previous version of the introduction talked about small bowel microbiota and antibiotic
resistance. The part about antibiotic resistance is now a separate part in the paper. By changing the title of
the paper, we believe that there is no need to have a separate section entitled “Small bowel microbiota”.
Point 7. From my point of view, this review is not a “Metadata Analysis” conducted according the Prisma
guidelines; the manuscript is a “Descriptive Review”; anyhow, authors should enquire the author´s
guidelines of MDPI Journal “Diseases” to clarify whether the section of “Material & Methods” has to be
included as seen in the current version.
Response 7: This section is excluded from the new version of the paper.
Point 8. Although in the title “3. Quorum sensing and biofilms…” indicates “in the small bowel” the content
of this section refers only “Gastrointestinal tract” so this should be clarified please.
Response 8: The title is now changed to “Quorum sensing and biofilms in the gastrointestinal tract”.
Point 9: In the section “4. Production of neurotransmitters by gut microbiota” clarify please whether is in
the “Gastrointestinal tract” or specifically in “Small bowel”
Response 9: The title is now changed to “Production of neurotransmitters by gut in the gastrointestinal
tract”.
Point 10: it would be very valuable for the readers to include a Table in the section “5. Small intestine
Bacterial Overgrowth…” that recapitulates main findings of experimental assays and studies based on
clinical trials and potential mechanisms underlie SIBO
Response 10: This Table is now included in the new version.
Point 11: it would be very valuable for the readers to include a Figure that depicts mechanisms underlie
SIBO based on the aforementioned Table.
Response 11: The title is no longer called “Small Intestine Bacterial Overgrowth Inflammation and Cancer”.
Because of this, we believe that having both the Table and Figure about SIBO would highlight SIBO too
much.
Point 12: In the sections “5.1 Celiac Disease and the gut microbiota”, “5.2 Autism spectrum disorder and
gut microbiota” and “5.3 Crohn´s disease and gut microbiota” clarify please explicitly whether these
diseases affect the small bowel or the whole gastrointestinal tract
Response 12: These diseases affect the whole gastrointestinal tract. This can now be concluded from the
new title.
Point 13: The conclusion should highlight the impact of SIBO in diagnosis and potential therapeutic
interventions of pharmaceutical and/immunologic type that reduce and/or replace the use of antibiotics
Response 13: This is now changed.
Point 14: check out and correct please “Chron´s disease” spelling in the body text and key words
See please in attached PDF additional corrections
Response 14: This is now changed.

Round 2

Reviewer 2 Report

Diseases-1721881

Title: Gastrointestinal Tract Microbiome Effect and Role in Disease Development

Authors addressed most concerns raised in the first peer-review round, and some points remain for correcting and/addressing in this manuscriptR2 version

Abstract: should be congruent with the title and the content of this review

Introduction

Page 2 Regarding the enunciates "Because..gut microbiota." can be merged due the content  is repeated on page 3 third paragraph "Studying...associated diseases."

Page 3 about the aim, it is important to underlie the originality in regard previous manuscripts; clarify please whether a special emphasis on the small bowel was placed

Page 3 about the heading “Impact of antibiotic resistance on small bowel environment does not reflect the content of this section, clarify please

Page 7 about the heading “Celiac disease and the gut microbiota” clarify please whether it refers "Celiac disease and the small bowel microbiota”?

Author Response

Point 1: Abstract should be congruent with the title and the content of this review.
We wrote a new abstract that fits the title better. Also, some changes were made in the conclusion in
order for it to be be congruent with the title.
Point 2: Introduction- Page 2 Regarding the enunciates "Because..gut microbiota." can be merged due
the content is repeated on Page 3 third paragraph "Studying...associated diseases."
This is now corrected.
Point 3: Page 3 about the aim, it is important to underlie the originality in regard previous manuscripts;
clarify please whether a special emphasis on the small bowel was placed.
There is no special emphasis on the small bowel.
Point 4: Page 3 about the heading “Impact of antibiotic resistance on small bowel environment” does
not reflect the content of this section, clarify please.
The title of this heading is now changed to “Impact of antibiotic resistance on gastrointestinal tract
environment”. We believe that additional changes in the paragraph are not needed due to the fact that
the context would be lost.
Point 4: Page 7 about the heading “Celiac disease and the gut microbiota” clarify please whether it
refers "Celiac disease and the small bowel microbiota”?
This part of the paper refers to the gastrointestinal tract. This is mentioned in the paragraph. 
